# MAGNET: Counterfactual samples synthesizing for mitigating hallucination in large language models

**Byeong Su Kim** [1,2], **Beomsoo Kim** [3]*, **Beakcheol Jang** [3]*

**1** IKLAB Inc., Geumcheon-gu, Seoul, South Korea, **2** Department of Artificial Intelligence, Yonsei University, Seodaemun-gu, Seoul, South Korea, **3** Graduate School of Information, Yonsei University, Seodaemun-gu, Seoul, South Korea

* beomsoo@yonsei.ac.kr (BK); bjang@yonsei.ac.kr (BJ)

## Abstract

Hallucinations are widely recognized as a significant drawback of large language models. Several attempts have been made to reduce the intensity of hallucinations. Among the various attempts, our research has been directed towards mitigating hallucinations caused by the co-occurrence statistics of pre-training corpora. We introduce Model-AGNostic countErfacTual synthesis and adaptive fine-tuning framework (MAGNET), a fine-tuning method that can mitigate the bias of co-occurrence statistics on large language models pre-training data when generating sentences. Our pipeline generates the counterfactual sample sentences and subject and object information for the counterfactual sample from the language model, and filters them to make sure they contain these three pieces of information before using them as fine-tuning data. Next, it utilizes both the generated counterfactual sample and the original sentence used to generate it as a training dataset. When our method is applied to GPT-Neo 2.7B model, it shows a 12% improvement in the Factual Knowledge Probing experiment, and there is a correlation analysis that can mitigate the bias on the pre-training data. In the TruthfulQA experiment, when fine-tuning the GPT-Neo 125M model on the LAMA-TREx dataset, applying our method showed 2.27% better performance than not applying it.

## Introduction

Natural language processing (NLP) research has recently experienced rapid growth with the emergence of large language models (LLMs) [1,2]. LLMs have demonstrated strong performance across a wide range of NLP tasks, including natural language inference [3], question answering [4], common-sense reasoning [5], and translation [6]. They have also achieved significant gains in natural language generation tasks. However, the problem of hallucination—the generation of plausible but untruthful sentences—has attracted considerable attention. Early work focused on the likelihood-maximizing objective function used during training and decoding, showing

**Data availability statement:** All dataset files used in this study is publicly available at https://doi.org/10.6084/m9.figshare.30304348.v1.

**Funding:** This work was supported by the Ministry of Education of the Republic of Korea and the National Research Foundation of Korea (NRF-2024S1A5C3A03046579). The funders had no role in study design, data collection and analysis, decision to publish, or preparation of the manuscript.

**Competing interests:** The authors have declared that no competing interests exist.

that natural language generation models can produce sentences that are plausible yet nonsensical or untruthful [7,8].

Recent studies suggest that LLMs often learn spurious features, which can lead to untruthful sentences [9]. Inspired by [10], we identify co-occurrence statistics in pre-trained sentences as a major contributor to these spurious features. Kang et al. proposed a fine-tuning method that removes biased samples from the dataset. While this approach mitigates hallucination caused by high co-occurrence statistics, it can hurt generalization due to the reduced data size.

In this paper, we propose MAGNET (Model-AGNostic coutErfacTual synthesis and adaptive fine-tuning), a framework designed to address bias in fine-tuning datasets by generating counterfactual samples for all instances, rather than removing biased samples. Counterfactual samples have been widely used in NLP to mitigate spurious features such as co-occurrence bias [11], and several studies have leveraged them for data augmentation [12–15]. Most methods generate counterfactuals by identifying and replacing terms that play a crucial role in a sentence's causality.

Using MAGNET presents two main challenges. First, generating counterfactuals to address subject-object co-occurrence bias requires extracting the subject and object, typically using part-of-speech (POS) tagging. In our approach, we directly utilize the subject and object information provided by LAMA-TREx. Second, counterfactual sentences should retain the subject while negating the object. This task requires broad knowledge and common-sense reasoning. To address this, we leverage GPT-3's powerful few-shot learning ability to generate counterfactual sentences effectively.

## Related works

### Spurious features in language models

LLM often produces plausible sentences that have no basis in truth [8,16]; this is because LLM learns shortcuts by relying on spurious features when learning, and spurious features include word-overlap, priming, surface form, and co-occurrence [10,17–19].

Word-overlap is one of the shortcuts that predicts the answer to entailment, contradiction, or neutrality from the perspective of natural language inference by learning the frequency of words commonly used between premise and hypothesis. For example, "The doctors visited the lawyer" as a premise and "The lawyer visited the doctors" as a hypothesis is a non-entailment sentence. However, LLM, which has learned the word-overlap bias, judges it as an entailment sentence.

Priming is an unconscious form of human memory that involves the perceptual identification of words and objects [20]. It refers to the pre-contextual effect that influences the interpretation of new or unfamiliar information. For example, if you are asked to fill in the blank in 'SO_P' and you have recently heard the word 'eat,' you are more likely to complete the word with 'SOUP,' whereas if you have just come out of the bath, you are more likely to complete the word with 'SOAP.'

Surface form refers to relying too heavily on the surface form of an entity name, such as predicting that a person with an Italian-pronounced name will speak Italian, regardless of the facts.

Co-occurrence implies that the subject and object occur simultaneously in the pre-training data. For example, if the pre-training data contains a total of 1,337,774 sentences with the subject 'Texas' and the object 'Houston' and 1,217,494 sentences with the subject 'Texas' and the object 'Austin', a model pre-trained on this data might output the word Houston when the sentence 'The capital of Texas is' is input, even though Austin is the correct answer.

These spurious features can help generate plausible sentences but are not suitable for generating factual sentences. In addition, existing evaluations have not been able to control these spurious features; therefore, new evaluation methods have been proposed [21]. The study [22] also found that removing spurious features reduced the accuracy of the model. In our study, we created a counterfactual sample to avoid overparameterization owing to spurious feature removal.

## Counterfactual data augmentation

Recently, augmenting counterfactual data has emerged as a way to mitigate spurious correlations and increase model robustness [23]. [24] employed humans to augment counterfactual examples. They found that counterfactually generated data mitigated spurious patterns in the training data. However, these methods are expensive, time-consuming, and prone to human error. In contrast, there are two main methods for automatic generation: 1) rule-based methods using certain templates or patterns and 2) deep learning-based language models.

Rule-based methods include [25], which uses templates, and [26], which uses decision trees. Owing to the well-defined rules, this method produces well-balanced sentences as intended by the authors. However, owing to the complexity of the rules, the sentences generated are uncreative and too monotonous. In addition, rules may be followed too frequently, resulting in nonsensical sentences that are not applicable to the task. A recent study [27] proposed two ways to adjust the perturbation: adjusting the size of the word replacement in the sentence and adjusting the offset of the sentence matrix representation, to generate richer sentences.

There have also been attempts to use deep learning-based language models to generate counterfactual samples. One such example is Polyjuice [28]. Polyjuice combines a finetuned GPT-2 [29] model with control codes to generate a variety of sentences that match the control codes. [30] used LaMDA [31] to generate counterfactual samples; these are then subjected to human ratings to get a high-quality, diverse, and complex sample.

To generate counterfactual samples for common sense, we adopt the method of using deep learning-based language models to fully utilize the knowledge retrieval and reasoning capabilities of LLMs. Prompting is scalable because it allows pre-trained models to adapt well to various tasks and domains without parameter modification. LLMs such as GPT-3 [32] have shown strong zero-shot and few-shot performance with prompting.

## Prompt tuning and fine-tuning

To improve the performance of LLMs, there have been two main methods, prompt tuning [33–38] and fine tuning, and GPT-3, in particular, has shown the possibility of solving various tasks with zero or few shot methods. However, manually writing prompts is not a simple task, and the proposed Mining-based Prompt tuning [39,40] and Learning-based Prompt Tuning [41,42] methods require prompt data that can be extracted and ranked, or learning additional models to rewrite prompts. [43] explains that finetuned LMs perform better at factual knowledge probing than prompt-tuned LMs, and while GPT-3 and T0 were designed to perform well on a variety of tasks without fine-tuning [32,44], recent research has shown that fine-tuning LLMs improves performance on reasoning [45], report generation [46], and more.

## Materials and methods

### Experimental setup

**Target model.** We used GPT-Neo 125M, GPT-Neo 1.3B, and GPT-Neo 2.7B, which are open-source versions of GPT-3. The model is publicly available at Huggingface's transformers [47]. The model is pre-trained on The Pile dataset. The Pile [48] is an open-source language modelling dataset that combines 22 small and high-quality datasets.

**Training data synthesis details.** To generate the counterfactual sentences, we used GPT-3.5 Turbo and 10 in-context examples of common common sense. The samples were human-written and followed the rules of retaining the subject but negated the object. We mainly sample counterfactual sentences for biased examples, such as grass and green, animal and dog, which tend to co-occur in general.

From this generated sample, we formally checked that Counterfactual, masked counterfactual, and [MASK] generated the three items well.

## Methods

In this section, we introduce MAGNET, a Model-AGNostic countErfacTual synthesis and adaptive fine-tuning framework. Fig 1 is an overview of MAGNET.

MAGNET comprises two main steps:

1. Synthesize a counterfactual sample for the training dataset.
2. Adaptive fine-tuning was performed on the data in the counterfactual sample and the corresponding existing training dataset.

In the following sections, we introduce how to synthesize counterfactuals of training data using GPT and adaptively fine-tune the language model between the generated data and the original data.

**Training data synthesis.** The rule that the counterfactual sample in this study must follow is to negate the object while maintaining the subject.

We chose LAMA-TREx because it contains the subject and object information needed to comply with this rule and can provide the appropriate information for the prompts needed to synthesize the sample.

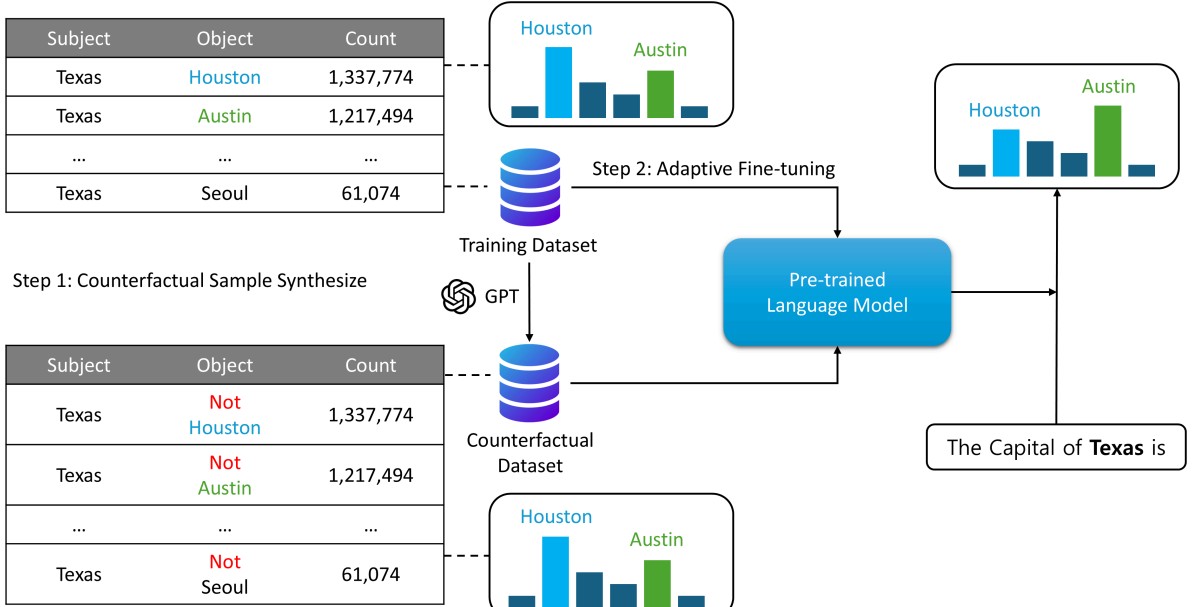

**Fig 1**. **Proposed framework: Counterfactual generation and bias reduction.**

In addition, we used GPT-3 to generate counterfactual samples. Existing methods for performing knowledge-based related tasks retrieved external knowledge from various sources of knowledge graphs [49–51], Wikipedia [52,53] and web search [54,55]

However, a recent study [56] shows that LLMs such as GPT-3 are particularly efficient in text-generation tasks; this is due to the LLM's superior knowledge retrieval and reasoning capabilities.

We go through the same process as in Fig 2 to synthesize fine-tuning data for the language model to mitigate bias in the co-occurrence statistics. The prompt $P$ provided to the GPT (the full content of which is provided in S1 File) consists of a Task instruction $T$, In-context Examples $I$, and Example to synthesize $E$:

$$P = (T, I, E) \tag{1}$$

where $T$ describes the rules that the counterfactual sample should obey and $I = \{i_1, \ldots, i_{10}\}$, which contains examples that generate counterfactuals for human-constructed commonsense.

Finally, the $E$ consists of a sentence $S$ to generate a counterfactual, a subject $Sub$ for the sentence, and an object $Obj$ for the sentence:

$$E = (S, Sub, Obj) \tag{2}$$

In the LAMA-TREx dataset, there are sentences for each fact, and these sentences are composed of *masked_sentence* with the object processed as [MASK], *sub_surface*, the subject information for the *masked_sentence*, and *obj_surface*, the object information.

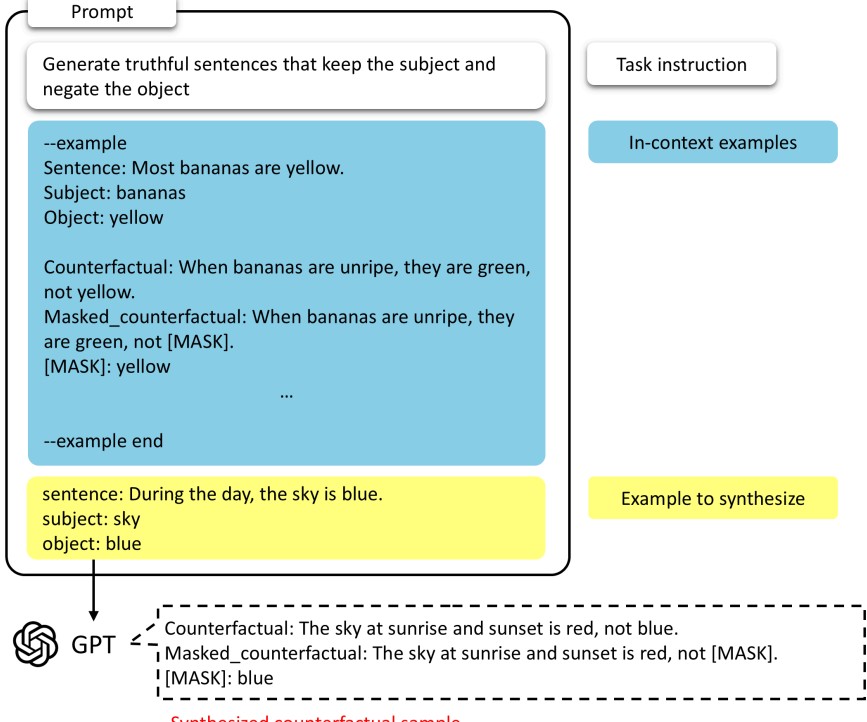

**Fig 2. GPT in-context learning for counterfactual sample generation.**

**Language model adaptive fine-tuning.** In our study, we performed adaptive fine-tuning to adjust the model for sentences based on triple-data to remove the bias for co-occurrence.

Instead of commonly used fine-tuning, for example, supervised fine-tuning for natural language inference and classification tasks, we reuse the next word prediction, which is an unsupervised pre-training already used for learning GPT.

Given a corpus of tokens $K = \{k_1, \ldots, k_n\}$, we use a standard language modeling objective to maximize the following likelihood:

$$L(k) = \sum_i \log P(k_i | k_{i-w}, \ldots, k_{i-1}; \Theta) \tag{3}$$

where $w$ is the context window size, i.e., the number of previous tokens that can be seen. And the conditional probability $P$ is modeled by a neural network with parameters $\Theta$. These parameters were learned using stochastic gradient descent [57].

GPT-Neo, which was used in our experiments, has the structure of a multi-layer transformer decoder because it is a large language model trained by exploiting the structure of the GPT. The model performs multi-head self-attention operations on input context tokens and applies a position-wise feedforward layer to generate output distributions over the target tokens:

$$
\begin{aligned}
h_0 &= KW_e + W_p \\
h_l &= \text{transformer\_block}(h_{l-1}) \forall i \in [1, n] \\
P(k) &= \text{softmax}(h_n W_e^T)
\end{aligned}
\tag{4}
$$

where $K = (k_{-w}, \ldots, k_{-1})$ is the context vector for the tokens, $n$ is the number of decoder layers, $W_e$ is the token embedding matrix, and $W_p$ is the position embedding matrix. The architecture of GPT-Neo is shown in Fig 3.

## Results

Table 1 provides a concise overview of the datasets, benchmarks, and evaluation metrics used in our experiments. Each experiment category—Factual Knowledge Probing, Counterfactual Training (MAGNET), Bias Analysis, and General Evaluation—is associated with its respective dataset and metrics, offering a clear summary of the experimental configuration prior to discussing detailed results.

### Factual knowledge probing

Fig 4 shows the results of the Factual Knowledge Probing experiment in the study by [10], which investigates the factual knowledge of LLMs using the LAMA-TREx dataset. The sentences used for validation are represented as subject-relation-object triples and converted into natural language using a predefined template. For example, the triple 'Texas'-'capital'-'Austin' is converted to "The capital of Texas is Austin." Each fact masks the object and is converted into a Cloze statement (e.g., "The capital of Texas is [MASK]").

We trained the model for 3 epochs on 4 RTX 3090 GPUs. The batch size per device was 32, giving a total batch size of 128. The learning rate was 2e-5, and the Adam optimizer was used with $\beta_1 = 0.9$ and $\beta_2 = 0.999$.

For fine-tuning, the input prompt follows the format "### Input:\n {X} \n\n### Response:", where X is a masked sentence. For instance, "Hydatius has the position of [MASK]." The model is supervised to predict "bishop," which is the expected answer. Details are provided in S2 File.

The factual knowledge dataset contains 20,587 samples. We used 10,294 original sentences and 10,294 counterfactual samples as random samples to train MAGNET. To evaluate the quality of counterfactuals, we computed Self-BLEU, which measures sentence similarity. The score of 0.4668 indicates moderate diversity, showing that the generated sentences are sufficiently varied while remaining natural and coherent. This balance is important for effective fine-tuning.

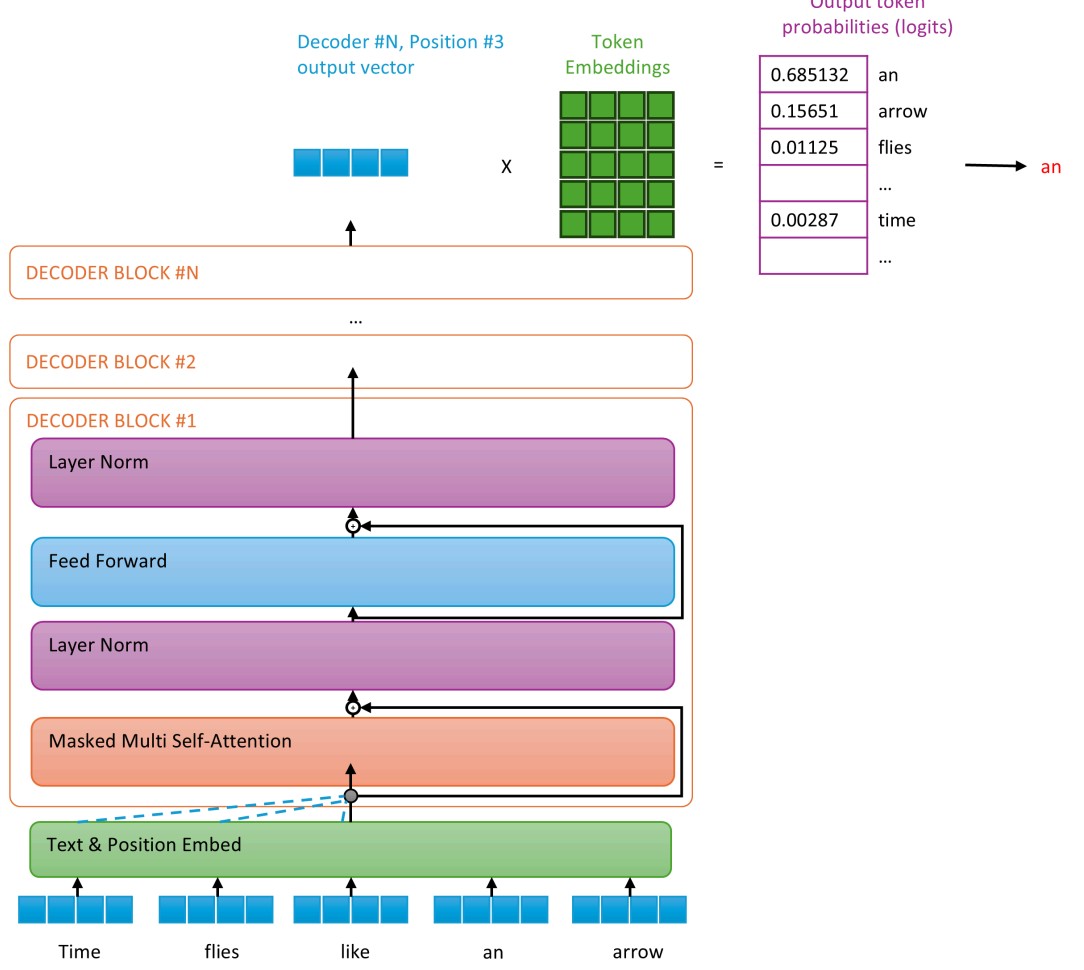

**Fig 3**. **GPT-Neo architecture for next-word prediction, similar to GPT-3.**

**Table 1**. **Overview of datasets, benchmarks, and corresponding evaluation metrics for each experiment category.**

| Category | Dataset / Benchmark | Details / Metrics |
|---|---|---|
| Factual Knowledge Probing | LAMA-TREx | 41 relations, 20,587 train / 8,824 test, evaluated with Hits@1 |
| Counterfactual Training (MAGNET) | Generated CS from LAMA-TREx | 10,294 original + 10,294 CS (Self-BLEU = 0.4668) |
| Bias Analysis | The Pile (GPT-Neo pre-training corpus) | Co-occurrence statistics of subject-object pairs |
| General Evaluation | TruthfulQA | MC2 score (0-shot) |
| | HellaSwag | Multiple-choice accuracy (10-shot) |
| | Winogrande | Multiple-choice accuracy (5-shot) |

For evaluation, we used Hits@1. It is 1 if the correct answer is ranked first among predicted candidates, and 0 otherwise. Because LLMs are not specifically trained for factual knowledge probing, we tested three restricted output vocabularies: (1) *remove stopwords*, (2) *gold objects*, and (3) *gold objects (relation-wise)*. The first excludes NLTK 3.8.1 stopwords. The second restricts candidates to gold objects in the entire dataset, while the third restricts them per relation.

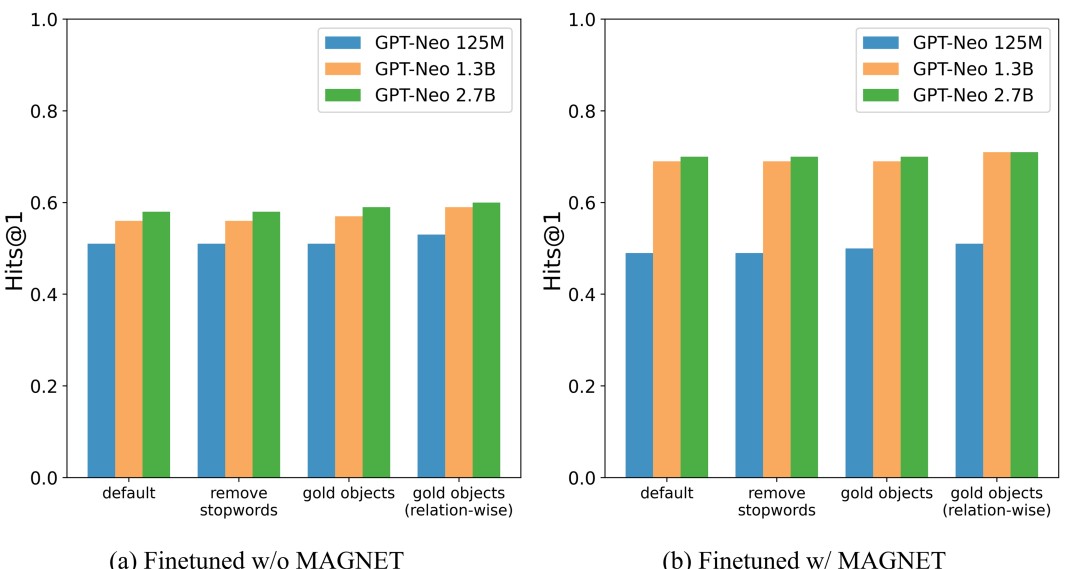

(a) Finetuned w/o MAGNET (b) Finetuned w/ MAGNET

**Fig 4**. **MAGNET effect on Hits@1: improves GPT-Neo 2.7B by 0.12 and 1.3B by 0.13; fine-tuning alone has minimal impact.**

Fig 5 shows Hits@1 under a zero-shot setting with limited candidate sets. MAGNET improves the score by 0.12 for the largest model and by 0.13 for the 1.3B model.

Table 2 compares GPT-Neo 2.7B performance under different training strategies. The Baseline model does not address subject-object co-occurrence biases, resulting in moderate Hits@1 scores. Undersampling removes biased samples, reducing training data and diversity. This increases overfitting risk and lowers generalization, especially in Gold Objects and Relation-wise evaluations. In contrast, MAGNET generates counterfactual samples that negate frequent object associations while preserving subjects. Learning from both original and counterfactual data maintains diversity and improves generalization, yielding substantially higher Hits@1 scores across all evaluation scenarios.

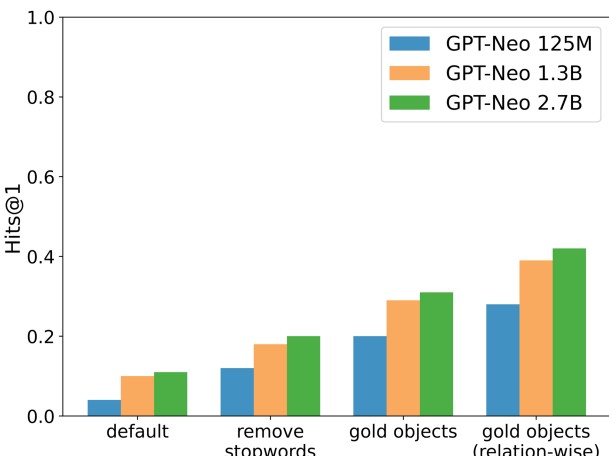

**Fig 5**. **Zero-shot Hits@1: larger models and more restricted candidates yield higher scores.**

**Table 2**. Comparison of Hits@1 performance on Factual Knowledge Probing across different training methods.

| Method | Hits@1 | Hits@1 (remove stopwords) | Hits@1 (Gold Objects) | Hits@1 (Relation-wise) |
|---|---|---|---|---|
| Baseline (w/o MAGNET) | 0.58 | 0.58 | 0.59 | 0.60 |
| Undersampling | 0.52 | 0.53 | 0.55 | 0.57 |
| MAGNET | 0.70 | 0.70 | 0.70 | 0.71 |

Baseline represents training without counterfactuals, Undersampling mitigates co-occurrence bias by removing biased samples, and MAGNET leverages counterfactual samples for fine-tuning. Scores are reported for various candidate restrictions: full set, stopwords removed, gold objects, and relation-wise gold objects.

## Correlation analysis

We analyzed co-occurrence statistics in the Pile dataset [48], a pre-training dataset for GPT-Neo, and correlated them with LLMs' ability to probe factual knowledge. Entities with uncountable co-occurrence counts or consisting of more than three tokens (less than 6% of all entities) were excluded. We then computed correlations for (1) zero-shot, (2) fine-tuning alone, and (3) fine-tuning using MAGNET.

Fig 6 illustrates the number of samples in each joint subject-object frequency bin, organized according to the subject frequency bin.

Fig 7 shows co-occurrence correlations in the zero-shot setting. Hits@1 scores increase linearly with subject frequency up to approximately $10^4$-$10^5$ for the joint subject-object frequency. However, for high-frequency subjects with relatively rare object occurrences, Hits@1 drops sharply. This indicates that LLMs struggle to predict rare facts due to co-occurrence bias.

Fig 8 presents co-occurrence correlations for fine-tuning and MAGNET. Fine-tuning roughly doubles Hits@1 compared to zero-shot but still shows sharp drops for rare facts. MAGNET, in contrast, improves overall performance by approximately three times over zero-shot and shows a slower performance decline, even for rare subject-object pairs. For

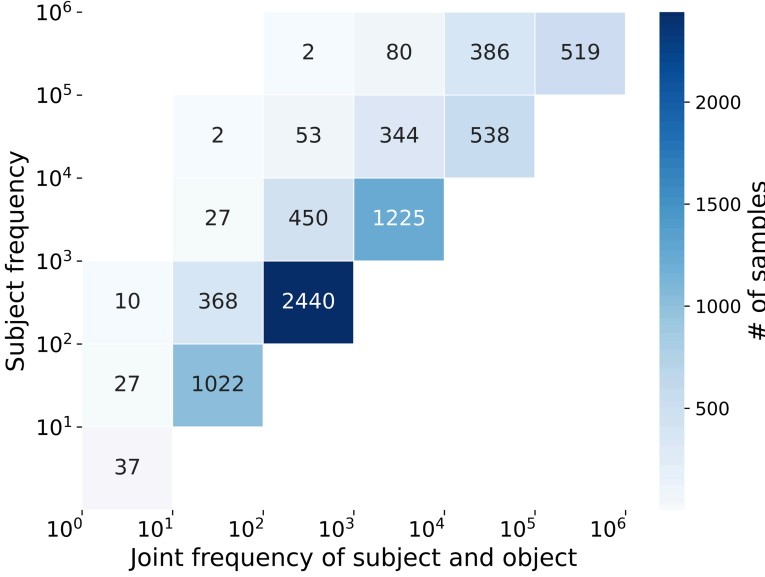

**Fig 6**. Subject and joint frequency analysis in pre-training data for factual knowledge probing outputs.

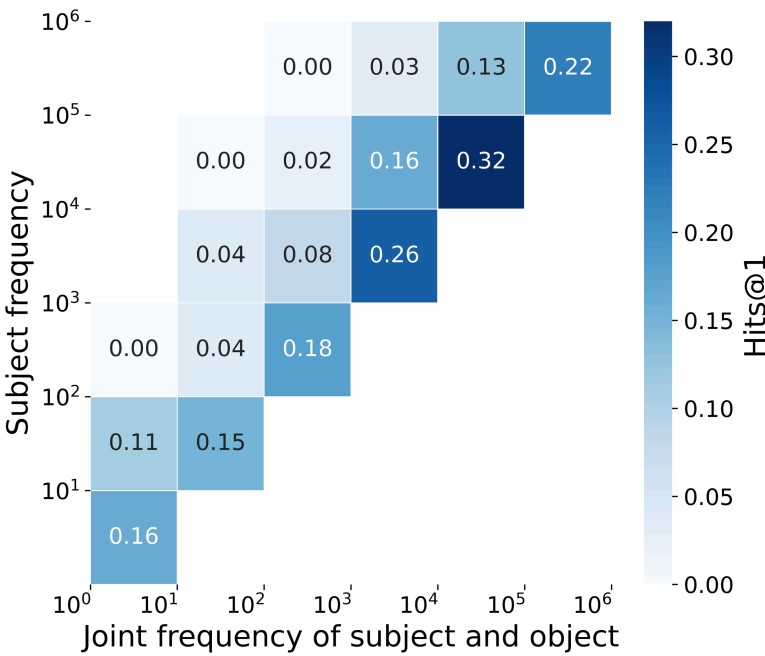

**Fig 7**. Correlation between the conditional probability of subject-object pairs and Hits@1 in GPT-Neo 2.7B pre-training under the *remove stopwords* setting.

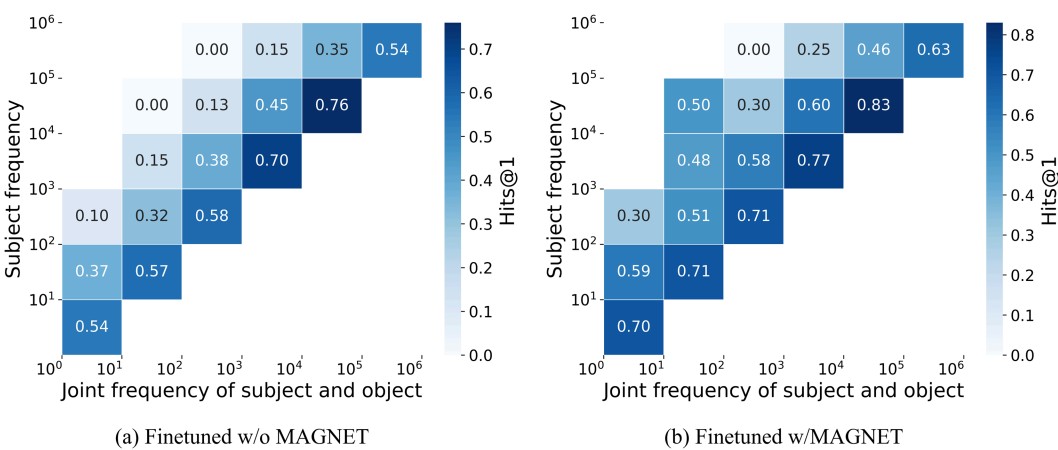

(a) Finetuned w/o MAGNET                    (b) Finetuned w/MAGNET

**Fig 8**. MAGNET's impact on Hits@1: improves performance over fine-tuning, especially for rare subject-object pairs, with GPT-Neo 2.7B pre-trained under the *remove stopwords* setting.

bins with subject frequency $10^3$-$10^4$ and joint frequency $10^1$-$10^2$, MAGNET demonstrates about threefold robustness to co-occurrence bias compared to zero-shot and slower decline than standard fine-tuning.

Without MAGNET, GPT-Neo 2.7B produced 5,154 correct and 3,670 incorrect answers out of 8,824. With MAGNET, the model achieved 6,177 correct and 2,647 incorrect answers. This means 1,521 predictions changed from incorrect to correct, while 498 changed from correct to incorrect.

Among the 3,670 errors without MAGNET, 989 cases involved predictions of words with higher co-occurrence counts than the ground truth. MAGNET corrected 311 of these bias-induced errors (examples in Table 3). Conversely, of the

**Table 3**. Examples where MAGNET corrected originally incorrect predictions.

| Query | Groundtruth | Prediction | |
|---|---|---|---|
| | | GPT-Neo 2.7B w/o MAGNET | GPT-Neo 2.7B w/ MAGNET |
| **Georgie Fame** plays | Piano (1130) | Guitar (1660) | Piano (1130) |
| **Unscripted** was originally aired on | HBO (3830) | Netflix (5030) | HBO (3830) |
| **Norite** is named after | Norway (80) | Nickel (138) | Norway (80) |
| **Deborah Wiles** was born in | Mobile (18) | Birmingham (26) | Mobile (18) |
| **Oliver Ellsworth** died in | Windsor (202) | Chicago (266) | Windsor (202) |

Examples where MAGNET corrected originally incorrect predictions. The table shows queries where GPT-Neo 2.7B originally predicted a word with higher co-occurrence than the ground truth, and how MAGNET corrected these predictions.

498 predictions that changed from correct to incorrect, 382 had higher conditional probabilities under the base model, indicating that MAGNET occasionally flipped answers despite the model's original preference for the correct option (examples in Table 4).

Overall, MAGNET effectively corrects bias-induced errors, though it occasionally flips correct answers to incorrect ones. These cases typically occur when the object distribution for a subject is relatively uniform, meaning no single object dominates co-occurrence statistics. As a future direction, constraining counterfactual generation to subjects with strongly skewed object distributions could reduce unnecessary flips and further improve model performance.

## Results for open LLM

We evaluated the impact of MAGNET on the target models across multiple benchmarks. In addition to TruthfulQA, we included HellaSwag and Winogrande, with results summarized in Table 5. For TruthfulQA, we used MC2 (Multi-true), which computes the normalized probability assigned to the correct answer set given multiple true/false options. HellaSwag and Winogrande were evaluated using multiple-choice accuracy, representing the proportion of correct selections among four candidate continuations and pronoun disambiguation questions, respectively.

Models were trained on 4 RTX 3090 GPUs for 3 epochs, using a batch size of 256 and a learning rate of 2e-5. The Adam optimizer was employed with $\beta_1 = 0.9$ and $\beta_2 = 0.999$. All other procedures follow HuggingFace's causal language modeling scripts [58]. Details of fine-tuning and evaluation are provided in S3 File.

We further investigated the effect of training data size on GPT-Neo 125M using MAGNET, as shown in Fig 9. These experiments were single runs.

Overall, MAGNET effectively mitigates co-occurrence bias, reducing the likelihood of generating incorrect words with high co-occurrence probability. This improves the factual accuracy and truthfulness of model outputs.

## Ablation study

To evaluate the quality of counterfactual sentences, we conducted generation experiments using 1-shot, 5-shot, and 10-shot prompt settings. Results are summarized in Table 6.

**Table 4**. Examples where MAGNET flipped originally correct answers into incorrect ones (predictions with similar co-occurrence likelihoods).

| Query | Groundtruth | Prediction | |
|---|---|---|---|
| | | GPT-Neo 2.7B w/o MAGNET | GPT-Neo 2.7B w/ MAGNET |
| **The Great Wall** is located in | China (1813) | China (1813) | Mongolia (1803) |
| **Shakespeare** wrote | Hamlet (177) | Hamlet (177) | Macbeth (169) |
| **Jupiter** is known as the largest | planet (19) | planet (19) | Saturn (18) |

Examples where MAGNET changed originally correct predictions to incorrect ones, but the alternative predictions had similar co-occurrence likelihoods, illustrating potential side-effects of bias correction.

**Table 5**. **Evaluation of MAGNET's generation performance compared to others.**

| Model | Model size (Parameters) | TruthfulQA (0-shot) | HellaSwag (10-shot) | Winogrande (5-shot) |
|---|---|---|---|---|
| gemma-2b-it | 2B | 45.82 | 62.7 | 60.93 |
| gemma-7b-it | 7B | 47.29 | 71.96 | 67.96 |
| GritLM-7B | 7B | 45.81 | 80.91 | 77.82 |
| Llama3-ChatQA-1.5-70B | 70B | 46.96 | **88.19** | **82.87** |
| GPT-Neo 125M | 125M | 45.58 | 30.4 | 50.43 |
| w/o MAGNET | | 45.67 ± 0.15 | 28.59 ± 0.21 | 49.88 ± 0.12 |
| w/ MAGNET | | **47.85** ± 0.13 | 29.43 ± 0.18 | 51.07 ± 0.10 |
| GPT-Neo 1.3B | 1.3B | 39.61 | 48.93 | 54.93 |
| w/o MAGNET | | 39.31 ± 0.12 | 46.75 ± 0.15 | 53.48 ± 0.10 |
| w/ MAGNET | | 40.30 ± 0.10 | 47.05 ± 0.14 | 54.14 ± 0.10 |
| GPT-Neo 2.7B | 2.7B | 39.86 | 55.80 | 57.70 |
| w/o MAGNET | | 40.86 ± 0.09 | 51.34 ± 0.11 | 56.03 ± 0.08 |
| w/ MAGNET | | 42.06 ± 0.09 | 52.73 ± 0.10 | 56.99 ± 0.07 |
| Llama-2 7B | 7B | 38.76 | 78.59 | 74.03 |
| w/o MAGNET | | 39.03 ± 0.06 | 75.48 ± 0.08 | 73.12 ± 0.05 |
| w/ MAGNET | | 40.01 ± 0.05 | 76.89 ± 0.06 | 73.55 ± 0.04 |

Evaluation of the generation performance of MAGNET and other models. Each model was evaluated over 5 independent runs; values are reported as mean ± 95% confidence interval (CI). Bold text indicates the highest score.

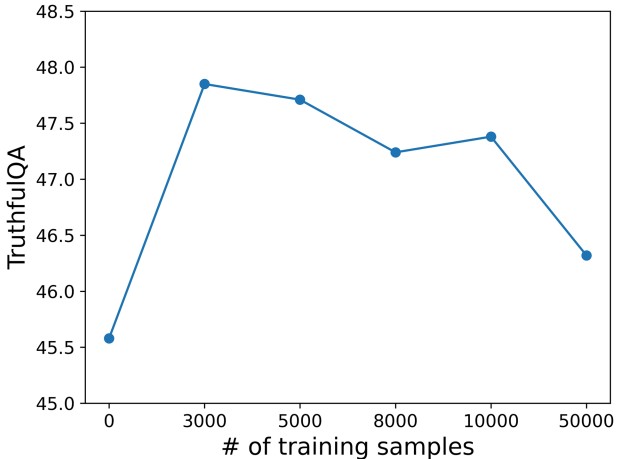

**Fig 9**. **Relationship between the number of GPT-Neo 125M fine-tuning samples and TruthfulQA scores, using MAGNET.**

**Table 6**. **Evaluation of counterfactual sentence generation performance across different n-shot prompts.**

| Model | n-shot | TruthfulQA | HellaSwag | Winogrande |
|---|---|---|---|---|
| GPT-Neo 125M | 1-shot | 46.10 ± 0.10 | 28.40 ± 0.13 | 50.10 ± 0.11 |
| GPT-Neo 125M | 5-shot | 46.80 ± 0.09 | 29.10 ± 0.12 | 50.50 ± 0.10 |
| GPT-Neo 125M | 10-shot | **47.85** ± 0.13 | **29.43** ± 0.18 | **51.07** ± 0.10 |

Values are reported as mean ± 95% confidence interval (CI) computed over 5 independent runs.

As Table 6 shows, increasing the number of shots improves the model's ability to follow the prompt and produce correctly formatted outputs, enabling more effective data collection. The 10-shot setting consistently yielded the highest performance across benchmarks.

To ensure the factual accuracy of generated sentences, we performed human filtering to verify that the subject and object were preserved. Table 7 compares performance with and without this filtering.

Human filtering consistently improved performance, confirming that maintaining subject-object fidelity and factual consistency enhances model outputs.

We also compared counterfactual generation using GPT-3.5 Turbo and GPT-4 Turbo. Table 8 summarizes the results.

As shown, GPT-4 Turbo consistently outperformed GPT-3.5 Turbo across all benchmarks. These results emphasize the importance of generating high-quality, factually grounded counterfactual sentences while preserving the subject-object structure. They also indicate that using a sufficient number of in-context shots and leveraging more advanced LLMs can further enhance MAGNET's effectiveness, improving both the factual robustness and truthfulness of the target models.

## Discussion

The experimental findings indicate that MAGNET substantially improves the factual robustness and truthfulness of LLMs by addressing biases introduced by co-occurrence patterns in the pre-training data. In the Factual Knowledge Probing task, MAGNET fine-tuning resulted in a notable increase in Hits@1 accuracy across model sizes, particularly showing a 12% improvement in the GPT-Neo 2.7B model. In the TruthfulQA benchmark, which evaluates the truthfulness of generative responses, MAGNET consistently outperformed baseline models, with a maximum improvement of 2.27% in the GPT-Neo 125M setting.

These results validate our hypothesis that hallucinations in LLMs often stem from spurious correlations, particularly co-occurrence biases between subjects and objects in pre-training corpora. Traditional mitigation strategies—such as filtering out biased samples—tend to reduce data volume and hurt generalization performance. In contrast, MAGNET synthesizes counterfactual samples that negate the object while preserving the subject, offering a more data-efficient and scalable approach. This allows models to encounter alternative semantic structures during training without sacrificing data diversity.

Importantly, our analysis of the co-occurrence frequency in the Pile dataset revealed that LLMs tend to over-predict high-frequency object associations, even when they are incorrect. By introducing counterfactual samples that deliberately break these associations, MAGNET enables the model to better distinguish between frequency-based and fact-based predictions. This effect was most pronounced in low-frequency subject-object pairs, where traditional models performed

**Table 7**. **Evaluation of counterfactual sentence generation performance across different n-shot prompts, with and without human filtering.**

| Model | n-shot | TruthfulQA | HellaSwag | Winogrande |
|---|---|---|---|---|
| GPT-Neo 125M | 1-shot, w/o filtering | 46.10 ± 0.10 | 28.40 ± 0.13 | 50.10 ± 0.11 |
| | 1-shot, w/ filtering | 46.50 ± 0.09 | 28.70 ± 0.12 | 50.40 ± 0.10 |
| GPT-Neo 125M | 5-shot, w/o filtering | 46.80 ± 0.09 | 29.10 ± 0.12 | 50.50 ± 0.10 |
| | 5-shot, w/ filtering | 47.10 ± 0.08 | 29.30 ± 0.11 | 50.80 ± 0.09 |
| GPT-Neo 125M | 10-shot, w/o filtering | 47.85 ± 0.13 | 29.43 ± 0.18 | 51.07 ± 0.10 |
| | 10-shot, w/ filtering | **48.10** ± 0.07 | **29.60** ± 0.08 | **51.30** ± 0.09 |

Values are reported as mean ± 95% confidence interval (CI) computed over 5 independent runs.

**Table 8**. **Comparison of counterfactual sentence generation performance using GPT-3.5 Turbo and GPT-4 Turbo across benchmarks.**

| Model | TruthfulQA | HellaSwag | Winogrande |
|---|---|---|---|
| GPT-3.5 Turbo | 47.85 ± 0.13 | 29.43 ± 0.18 | 51.07 ± 0.10 |
| GPT-4 Turbo | **49.11** ± 0.10 | **31.12** ± 0.09 | **53.85** ± 0.10 |

Values are reported as mean ± 95% confidence interval (CI) computed over 5 independent runs.

poorly. With MAGNET, these rare factual associations were retained more accurately, suggesting enhanced model generalization and resistance to spurious correlations.

Nonetheless, the approach introduces certain trade-offs. A small portion of correctly predicted samples without MAGNET became incorrect after applying it. Our analysis indicates that this mainly occurred when the distribution of possible objects for a given subject was relatively uniform, i.e., no single object strongly dominated. In such cases, MAGNET sometimes flipped the prediction despite the ground truth having a higher conditional probability. Therefore, future improvements may benefit from mechanisms to dynamically weight or filter counterfactuals, especially in high-confidence cases.

In a broader context, MAGNET has implications for improving factual alignment in LLMs across tasks such as open-domain question answering, commonsense reasoning, and factual sentence generation. As the complexity and deployment scale of LLMs continue to grow, mitigating training-set-driven biases will become increasingly critical. MAGNET demonstrates a promising direction toward achieving this goal without compromising the scalability and efficiency of fine-tuning workflows.

Furthermore, extending MAGNET to more complex reasoning settings remains an important avenue for future work. For instance, multi-hop reasoning often requires chaining intermediate facts, where co-occurrence biases can propagate across steps. Integrating MAGNET with chain-of-thought prompting may help stabilize such reasoning by reducing spurious associations at each step. Similarly, in temporally sensitive tasks where factual correctness depends on evolving knowledge, combining MAGNET with retrieval-augmented generation (RAG) could ensure that counterfactual training remains aligned with up-to-date evidence. Together, these directions highlight the broader applicability of MAGNET beyond single-hop factual recall.

## Conclusion

In this study, we introduced MAGNET, a model-agnostic counterfactual data synthesis and fine-tuning framework designed to mitigate hallucination in LLMs by addressing co-occurrence bias in pre-training corpora. Unlike prior approaches that remove biased samples at the cost of data volume and generalization, MAGNET augments training data with synthetically generated counterfactual sentences that retain the subject but negate the object. This enables models to learn more robust, factually grounded representations.

Our experiments demonstrate that MAGNET significantly improves performance across two key benchmarks. On the Factual Knowledge Probing task, we observed up to a 12% increase in Hits@1 accuracy, while in the TruthfulQA benchmark, MAGNET led to a 2.27% improvement in truthfulness for the GPT-Neo 125M model. Furthermore, correlation analysis confirmed that MAGNET reduces the model's over-reliance on spurious co-occurrence patterns, particularly in low-frequency scenarios.

These results highlight MAGNET's potential as a general-purpose bias mitigation technique for enhancing the factual reliability of LLMs. Its compatibility with various model sizes and architectures, along with its minimal reliance on manual annotation, makes it a scalable and practical solution. Future work may explore extending this framework to other forms of bias and broader NLP tasks.

## Supporting information

**S1 File. Prompts used to generate counterfactual samples.**
(PDF)

**S2 File. Data description of factual knowledge probing experiments.**
(XLSX)

**S3 File. Description of datasets used with MAGNET for the target model.**
(XLSX)

## Acknowledgments

We gratefully acknowledge EleutherAI and Hugging Face, along with their open-source communities, for their contributions to the development and maintenance of the GPT-Neo models and the Transformers library. Their work provided essential infrastructure and resources that significantly supported the implementation and evaluation of our proposed framework.

## Author contributions

**Conceptualization:** Byeong Su Kim, Beomsoo Kim, Beakcheol Jang.

**Data curation:** Byeong Su Kim.

**Formal analysis:** Byeong Su Kim.

**Funding acquisition:** Beomsoo Kim.

**Investigation:** Beomsoo Kim.

**Methodology:** Byeong Su Kim.

**Supervision:** Beakcheol Jang.

**Visualization:** Byeong Su Kim.

**Writing – original draft:** Byeong Su Kim.

**Writing – review & editing:** Byeong Su Kim, Beomsoo Kim, Beakcheol Jang.

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
