## [Decision Letter · Decision Letter 0]

4 Sep 2025

PONE-D-25-17618

MAGNET: Counterfactual samples synthesizing for mitigating hallucination in large language models

PLOS ONE

Dear Dr. Jang,

Thank you for submitting your manuscript to PLOS ONE. After careful consideration, we feel that it has merit but does not fully meet PLOS ONE’s publication criteria as it currently stands. Therefore, we invite you to submit a revised version of the manuscript that addresses the points raised during the review process.

We look forward to receiving your revised manuscript.

Kind regards,

Sonia Vasconcelos, PhD

Academic Editor

PLOS ONE

Journal Requirements:

“This work was supported by the Ministry of Education of the Republic of Korea and the National Research Foundation of Korea (NRF-2024S1A5C3A03046579)”

Additional Editor Comments:

Each Reviewer highlighted specific points requiring attention, including the need for further details on prompt design and generation settings and for addressing the "circular contamination problem" referenced by Reviewer #3. Statistical testing is required to clarify the significance of your results, as noted especially by Reviewers #1 and #2.

Reviewer's Responses to Questions

**Comments to the Author**

1. Is the manuscript technically sound, and do the data support the conclusions?

Reviewer #1: Yes

Reviewer #2: Yes

Reviewer #3: Yes

2. Has the statistical analysis been performed appropriately and rigorously?

Reviewer #1: No

Reviewer #2: Yes

Reviewer #3: No

3. Have the authors made all data underlying the findings in their manuscript fully available?

Reviewer #1: Yes

Reviewer #2: Yes

Reviewer #3: Yes

4. Is the manuscript presented in an intelligible fashion and written in standard English?

Reviewer #1: No

Reviewer #2: Yes

Reviewer #3: Yes

5. Review Comments to the Author

Reviewer #1: This paper introduces MAGNET, a novel framework that uses counterfactual data to reduce hallucinations in LLMs by targeting co-occurrence bias. The method is innovative and shows promising improvements in factual accuracy and truthfulness. However, the manuscript lacks statistical significance testing, ablation studies, and deeper analysis of failure cases. Writing is clear but needs minor edits, and reproducibility would benefit from more details on prompt design and generation settings.

Recommendations: Add statistical tests (like p-values or confidence intervals) to show that the results are reliable. Do ablation studies to show how different prompts and data filtering affect the results. Look into cases where the method makes correct answers wrong, and explain why that happens. Fix grammar issues and make the writing clearer; also, make the tables easier to read. Share the exact prompts and settings used to generate the counterfactual data so others can repeat the work.

Reviewer #2: Summary:

The manuscript proposes the MAGNET framework, which generates counterfactual samples and jointly fine-tunes them with the original data to mitigate co-occurrence bias in pre-training corpora, thereby reducing hallucinations in large language models. The method yields consistent improvements across GPT-Neo model sizes on Factual Knowledge Probing and TruthfulQA, with particularly strong gains for low-frequency facts.

Strengths:

1. Clearly defined problem and novel methodological approach.

2. Consistent and substantial performance improvements across multiple model scales.

3. Data augmentation strategy preserves dataset size compared to filtering-based methods.

4. Transparent methodological details supporting reproducibility.

5. Insightful error analysis revealing both bias sources and correction mechanisms.

Weaknesses:

1. Some correct predictions are degraded due to overcorrection.

2. Evaluation is limited to specific benchmarks; broader generalization remains untested.

3. Lack of ablation studies to isolate the contributions of each pipeline component.

4. No quantitative assessment of the linguistic diversity of generated counterfactuals.

Comments:

1. Include ablation experiments to quantify the contribution of each component (extraction, generation, filtering).

2. Conduct statistical significance testing to confirm robustness of improvements.

3. Extend evaluation to additional open-domain QA or fact-checking datasets.

4. Discuss applicability to multi-hop reasoning and temporally sensitive facts.

5. Assess linguistic quality and diversity of counterfactuals to ensure naturalness.

6. Explore confidence-based weighting of counterfactuals to reduce overcorrection.

Reviewer #3: Major concerns:

- Circular contamination problem – the authors are using GPT-3 to generate counterfactual samples, but GPT-3 itself hallucinates, so essentially it’s like the models are trained on hallucinated "corrections." To be more specific, it MAGNET wants to negate “blue” in the figure 1 example, the underlying LLM needs to know when the sky is not blue, which also requires it’s knowledge to be correct.

- The choice of GPT-3 for counterfactual generation seems outdated for a 2025 submission - GPT-4 or newer models would be more relevant for practical applications.

- There is also no comparison to other finetuning methods that address hallucinations (eg. Ref [10]) at all.

There are also other minor issues:

- In the reported numbers more carefully, MAGNET corrected 311 incorrect predictions but caused 498 new errors from previously correct answers - a net loss of 187 correct responses. The claimed 12% improvement needs clearer explanation given this apparent contradiction.

- Some presentation issues detract from the work: the figures are difficult to read due to low resolution, the lengthy tables interrupt the narrative flow (perhaps better suited for appendices), and Figure 7 would be more informative if it directly showed Hits@1 improvements rather than raw scores.

While the core concept of using counterfactual samples to address co-occurrence bias has merit, these methodological and presentation concerns limit the paper's impact. So I’m suggesting a major revision.

6. PLOS authors have the option to publish the peer review history of their article (what does this mean?). If published, this will include your full peer review and any attached files.

Reviewer #1: **Yes:** Rajan Das Gupta

Reviewer #2: No

Reviewer #3: No

---

## [Author Response · Author response to Decision Letter 1]

8 Oct 2025

Reviewer #1 - Comment 1

Comment

Add statistical tests (like p-values or confidence intervals) to show that the results are reliable.

Response

We thank the reviewer for the valuable feedback regarding the statistical reliability of our results. In response, we have updated Tables 4, 5, 6, and 7 to include 95% confidence intervals (CI) for all reported metrics, computed over 5 independent runs. By providing these CIs, we quantify the variability in the results and demonstrate that the observed improvements with MAGNET and across different experimental settings are statistically reliable. The mean ± 95% CI format allows readers to assess the significance and robustness of the reported performance gains.

[Page 10-11](Table 4, Table 5, Table 6, Table 7) – Section Results

Reviewer #1 - Comment 2

Comment

Do ablation studies to show how different prompts and data filtering affect the results.

Response

We sincerely thank the reviewer for the helpful suggestion. To address this, we conducted ablation experiments to evaluate how different prompt settings and human filtering affect MAGNET’s performance. We compared 1-shot, 5-shot, and 10-shot prompts to investigate the impact of the number of in-context examples. Additionally, for the 10-shot setting, we applied human filtering to ensure that the generated counterfactual sentences preserved the subject-object pairs and were factually accurate.

The results indicate that increasing the number of shots improves performance, as the model can better follow the given examples. Furthermore, human filtering on the 10-shot setting consistently enhanced performance across all benchmarks, demonstrating the importance of maintaining factual correctness and the subject-object structure in counterfactual data for effective MAGNET fine-tuning.

[Page 10-11](Line 299–323) – Section Ablation Study of Results

Ablation Study

For assessing the quality of counterfactual sentences, we conducted generation 300 experiments with 1-shot, 5-shot, and 10-shot prompt settings.

Table 5 shows that as the number of shots increases, the model is better able to follow the given prompt and produce correctly formatted outputs, which allows for more effective data collection. Across the different n-shot settings, sentences generated with 10-shot prompts led to the best performance.

We conducted human filtering to ensure that the generated sentences preserved the subject and object while remaining factually consistent.

Table 6 shows that human filtering was performed to ensure that generated sentences preserved the subject and object and were factually accurate, which improved performance across all benchmarks.

Together with the n-shot setting, these results indicate that maintaining the subject and object and generating factually grounded sentences, as proposed in this study, contributes to higher model performance.

Table 7 summarizes the performance of MAGNET when using different LLMs for counterfactual generation. While our experiments primarily used the Turbo models, as is common in many practical settings, GPT-4 demonstrated improved performance across all benchmarks compared to GPT-3.5. This highlights the importance of high-quality counterfactual samples for effective MAGNET fine-tuning. Furthermore, it emphasizes that generating factually grounded sentences while preserving the subject-object structure, employing a sufficient number of in-context shots, and leveraging up-to-date, high-capacity language models are key factors for maximizing MAGNET’s effectiveness. These findings suggest that adopting newer models in future applications can further enhance the factual robustness and truthfulness of LLMs.

Reviewer #1 - Comment 3

Comment

Look into cases where the method makes correct answers wrong, and explain why that happens.

Response

We thank the reviewer for this insightful comment. To address this, we analyzed instances where MAGNET flipped originally correct predictions into incorrect ones in the Factual Knowledge Probing experiments. As shown in Table 2 and Table 3, these cases mainly occur when the distribution of possible objects for a given subject is relatively uniform—i.e., no single object strongly dominates. In such situations, even though the ground truth has a higher conditional probability, MAGNET occasionally changes the prediction due to the introduction of counterfactual samples. This highlights a trade-off of counterfactual training: while it improves robustness against spurious co-occurrence biases, it can slightly affect predictions when multiple plausible objects exist for a subject.

[Page 8-9](Line 262–280) – Section Correlation Analysis of Results

Correlation Analysis

Without MAGNET, GPT-Neo 2.7B produced 5,154 correct and 3,670 incorrect answers out of 8,824. With MAGNET, the model achieved 6,177 correct and 2,647 incorrect answers, meaning that 1,521 cases were changed from incorrect to correct, while 498 cases were changed from correct to incorrect.

Among the 3,670 errors without MAGNET, we identified 989 cases where the model predicted a word with a higher co-occurrence count than the ground truth. Of these bias-induced errors, 311 were corrected by MAGNET (examples in Table 2). Conversely, among the 498 answers that changed from correct to incorrect, 382 had higher conditional probability on the ground truth under the base model, indicating that MAGNET occasionally flipped answers despite the original model preferring the correct option (examples in Table 3).

Our analysis confirms that MAGNET effectively corrects bias-induced errors while occasionally flipping originally correct answers to incorrect ones. We observed that these “correct → incorrect” cases tend to occur when the distribution of potential objects for a given subject is relatively uniform, i.e., no single object dominates the co-occurrence statistics. As a future direction, we hypothesize that constraining counterfactual sentence generation to subjects whose objects exhibit a strong skew (above a certain threshold) could reduce unnecessary flipping and further improve overall model performance.

Reviewer #1 - Comment 4

Comment

Fix grammar issues and make the writing clearer; also, make the tables easier to read.

Response

We thank the reviewer for the suggestion. We have revised the manuscript to improve grammar and overall clarity. In addition, to enhance readability, Table 2—where readability was previously limited—has been replaced with Figure 6, which presents the information more clearly. Other tables have also been reformatted to improve accessibility and consistency.

[Page 8](Figure 6) – Section Correlation Analysis of Results

[Page 8-9](Line 262–280) – Section Correlation Analysis of Results

Correlation Analysis

Without MAGNET, GPT-Neo 2.7B produced 5,154 correct and 3,670 incorrect answers out of 8,824. With MAGNET, the model achieved 6,177 correct and 2,647 incorrect answers, meaning that 1,521 cases were changed from incorrect to correct, while 498 cases were changed from correct to incorrect.

Among the 3,670 errors without MAGNET, we identified 989 cases where the model predicted a word with a higher co-occurrence count than the ground truth. Of these bias-induced errors, 311 were corrected by MAGNET (examples in Table 2). Conversely, among the 498 answers that changed from correct to incorrect, 382 had higher conditional probability on the ground truth under the base model, indicating that MAGNET occasionally flipped answers despite the original model preferring the correct option (examples in Table 3).

Our analysis confirms that MAGNET effectively corrects bias-induced errors while occasionally flipping originally correct answers to incorrect ones. We observed that these “correct → incorrect” cases tend to occur when the distribution of potential objects for a given subject is relatively uniform, i.e., no single object dominates the co-occurrence statistics. As a future direction, we hypothesize that constraining counterfactual sentence generation to subjects whose objects exhibit a strong skew (above a certain threshold) could reduce unnecessary flipping and further improve overall model performance.

[Page 9](Figure 9)

Correlation between GPT-Neo 125M pre-training co-occurrence frequencies and Hits@1 scores with MAGNET, under the remove stopwords setting.

Reviewer #1 - Comment 5

Comment

Share the exact prompts and settings used to generate the counterfactual data so others can repeat the work.

Response

We thank the reviewer for the suggestion. The exact prompts and generation settings used for creating the counterfactual data are provided in S1 Appendix, allowing other researchers to reproduce our experiments.

[Page 13](Line 402) – S1 Appendix

S1 Appendix. Prompts used to generate counterfactual samples

Below is the complete prompt for generating counterfactual sentences with GPT-3. The task was to generate text, and we provided 10 examples written by humans. GPT-3 generates counterfactual sentences at the end of the prompt.

Generate truthful sentences that keep the subject and negate the object for a given sentence, following the example format.

--example

sentence: Most bananas are yellow.

subject: bananas

object: yellow

counterfactual: When bananas are unripe, they are green, not yellow.

masked_counterfactual: When bananas are unripe, they are green, not [MASK].

[MASK]: yellow

sentence: The largest city in Canada is Toronto.

subject: Canada

object: Toronto

counterfactual: The capital of Canada is Ottawa, not Toronto.

masked_counterfactual: The capital of Canada is Ottawa, not [MASK].

[MASK]: Toronto

sentence: The color of the rose is red.

subject: rose

object: red

counterfactual: Roses come in many colors, and they're not always red.

masked_counterfactual: Roses come in many colors, and they're not always [MASK].

[MASK]: red

sentence: The color of the grass is green.

subject: grass

object: green

counterfactual: The color of dry grass is brown, not green.

masked_counterfactual: The color of dry grass is brown, not [MASK].

[MASK]: green

sentence: Time heals all wounds.

subject: time

object: wounds

counterfactual: Time can't heal all wounds. you need the right remedy for the right situation.

masked_counterfactual: Time can't heal all [MASK]. you need the right remedy for the right situation.

[MASK]: wounds

sentence: Diabetic patients should not eat fruits.

subject: Diabetic

object: fruits

counterfactual: This doesn't mean that diabetics shouldn't eat fruit, but taking into account the sugar and fiber content of fruit can help balance the effects.

masked_counterfactual: This doesn't mean that diabetics shouldn't eat fruit, but taking into account the sugar and fiber content of [MASK] can help balance the effects.

[MASK]: fruit

sentence: Fire hydrants are, in most cases, red in color. The red color is visually striking and helps firefighters spot them in an emergency.

subject: Fire hydrant

object: red

counterfactual: Depending on your location, fire hydrants may not be red, so it's a good idea to check the regulations in your specific area.

masked_counterfactual: Depending on your location, fire hydrants may not be [MASK], so it's a good idea to check the regulations in your specific area.

[MASK]: red

sentence: During the day, the sky is blue.

subject: sky

object: blue

counterfactual: The sky at sunrise and sunset is red, not blue.

masked_counterfactual: The sky at sunrise and sunset is red, not [MASK].

[MASK]: blue

sentence: He is a great engineer.

subject: He

object: engineer

counterfactual: There are not only men, but also women among engineers.

masked_counterfactual: There are not only men, but also women among [MASK].

[MASK]: engineer

sentence: The most common animal kept as a pet is a dog.

subject: animal

object: dog

counterfactual: The most commonly raised animal for food is not a dog, but a chicken.

masked_counterfactual: The most commonly raised animal for food is not a [MASK], but a chicken.

[MASK]: dog

--example end

sentence: It covers the World War II in Europe on a grand strategic scale between 1939 and 1945.

subject: World War II

object: Europe

------Prompt Ends Here------

Below is the text generated by GPT

counterfactual: World War II did not take place only in Europe, but also in other regions around the world between 1939 and 1945.

masked_counterfactual: World War II did not take place only in [MASK], but also in other regions around the world between 1939 and 1945.

[MASK]: Europe

Reviewer #2 - Comment 1

Comment

Include ablation experiments to quantify the contribution of each component (extraction, generation, filtering).

Response

We thank the reviewer for the suggestion. We conducted ablation experiments to assess the contribution of each component in MAGNET: prompt shot count (1-shot, 5-shot, 10-shot), human filtering (applied to 10-shot), and the counterfactual generation model (GPT-3.5 Turbo vs. GPT-4 Turbo).

[Page 10-11](Line 299-323) – Section Ablation Study of Results

Ablation Study

For assessing the quality of counterfactual sentences, we conducted generation 300 experiments with 1-shot, 5-shot, and 10-shot prompt settings.

Table 5 shows that as the number of shots increases, the model is better able to follow the given prompt and produce correctly formatted outputs, which allows for more effective data collection. Across the different n-shot settings, sentences generated with 10-shot prompts led to the best performance.

We conducted human filtering to ensure that the generated sentences preserved the subject and object while remaining factually consistent.

Table 6 shows that human filtering was performed to ensure that generated sentences preserved the subject and object and were factually accurate, which improved performance across all benchmarks.

Together with the n-shot setting, these results indicate that maintaining the subject and object and generating factually grounded sentences, as proposed in this study, contributes to higher model performance.

Table 7 summarizes the performance of MAGNET when using different LLMs for counterfactual generation. While our experiments primarily used the Turbo models, as is common in many practical settings, GPT-4 demonstrated improved performance across all benchmarks compared to GPT-3.5. This highlights the importance of high-quality counterfactual samples for effective MAGNET fine-tuning. Furthermore, it emphasizes that generating factually grounded sentences while preserving the subject-object structure, employing a sufficient number of in-context shots, and leveraging up-to-date, high-capacity language models are key factors for maximizing MAGNET’s effectiveness. These findings suggest that adopting newer models in future applications can further enhance the factual robustness and truthfulness of LLMs.

Reviewer #2 - Comment 2

Comment

Conduct statistical significance testing to confirm robustness of improvements.

Response

We thank the reviewer for the valuable feedback regarding the statistical reliability of our results. In response, we have updated Tables 4, 5, 6, and 7 to include 95% confidence intervals (CI) for all reported metrics, computed over 5 independent runs. By providing these CIs, we quantify the variability in the results and demonstrate that the observed improvements with MAGNET and across different experimental settings are statistically reliable. The mean ± 95% CI format allows readers to assess the significance and robustness of the reported performance gains.

[Page 10-11](Table 4, Table 5, Table 6, Table 7) – Section Results

Reviewer #2 - Comment 3

Comment

Extend evaluation

---

## [Decision Letter · Decision Letter 1]

25 Nov 2025

PONE-D-25-17618R1MAGNET: Counterfactual samples synthesizing for mitigating hallucination in large language modelsPLOS ONE

Dear Dr. Jang,

Thank you for submitting your manuscript to PLOS ONE. After careful consideration, we feel that it has merit but does not fully meet PLOS ONE’s publication criteria as it currently stands. Therefore, we invite you to submit a revised version of the manuscript that addresses the points raised during the review process.

**Overall, all reviewers found that the revision addressed their comments. However, Reviewer#1 is satisfied but notes a few additional points that need your attention. Please address them before the final decision.**

We look forward to receiving your revised manuscript.

Kind regards,

Sonia Vasconcelos, PhD

Academic Editor

PLOS ONE

**Journal Requirements:**

Reviewers' comments:

Reviewer's Responses to Questions

**Comments to the Author**

1. If the authors have adequately addressed your comments raised in a previous round of review and you feel that this manuscript is now acceptable for publication, you may indicate that here to bypass the “Comments to the Author” section, enter your conflict of interest statement in the “Confidential to Editor” section, and submit your "Accept" recommendation.

Reviewer #1: All comments have been addressed

Reviewer #2: All comments have been addressed

Reviewer #3: All comments have been addressed

2. Is the manuscript technically sound, and do the data support the conclusions?

Reviewer #1: Yes

Reviewer #2: Yes

Reviewer #3: Yes

3. Has the statistical analysis been performed appropriately and rigorously?

Reviewer #1: Yes

Reviewer #2: N/A

Reviewer #3: Yes

4. Have the authors made all data underlying the findings in their manuscript fully available?

Reviewer #1: Yes

Reviewer #2: Yes

Reviewer #3: Yes

5. Is the manuscript presented in an intelligible fashion and written in standard English?

Reviewer #1: Yes

Reviewer #2: Yes

Reviewer #3: Yes

6. Review Comments to the Author

**Reviewer #1:** The manuscript is well revised and substantially improved. Only a few minor editorial adjustments are recommended before final acceptance:

1. Please ensure consistent formatting of performance metrics — for example, use “Hits@1” uniformly throughout the text, tables, and figures.

2. Some figure captions (especially Figures 7–9) could be made more concise, and figure quality should be upgraded to ensure all text and visual elements are clearly visible.

3. Including a short summary table in the Methods section listing datasets, benchmarks, and evaluation metrics would improve readability.

4. A light grammar and style edit is advised to smooth a few long sentences in the Introduction and Results sections.

5. Verify reference formatting and ensure that all appendices (S1–S3) are correctly labeled and cited in the main text.

These are minor refinements. Overall, the paper is strong, clearly presented, and ready for publication once these small adjustments are made.

**Reviewer #2:** (No Response)

**Reviewer #3:** The additional experiments and discussion addressed other reviewers' and my concerns well. So I think it's ready for acceptance.

7. PLOS authors have the option to publish the peer review history of their article (what does this mean?). If published, this will include your full peer review and any attached files.

Reviewer #1: No

Reviewer #2: No

Reviewer #3: No

---

## [Author Response · Author response to Decision Letter 2]

30 Nov 2025

Original Manuscript ID PONE-D-25-17618R1

Original Article Title MAGNET: Counterfactual samples synthesizing for mitigating hallucination in large language models

Reviewer #1 - Comment 1

Comment Please ensure consistent formatting of performance metrics — for example, use “Hits@1” uniformly throughout the text, tables, and figures.

Response We thank the reviewer for pointing out the inconsistency in the formatting of performance metrics. We have corrected the instance of “Hit@1” on line 212 to “Hits@1” and updated the caption of Figure 9, which was previously mislabeled. Consequently, the Results section now consistently uses the Hits@1 metric for Factual Knowledge Probing and Correlation Analysis, while the Results for Open LLM and Ablation Study sections report only the MC2 (Multi-true) and multiple-choice metrics, ensuring uniformity across the text, tables, and figure captions.

[Page 6-11](Line 181-297) – Section Results

Results

Table 1 provides a concise overview of the datasets, benchmarks, and evaluation metrics used in our experiments. Each experiment category—Factual Knowledge Probing, Counterfactual Training (MAGNET), Bias Analysis, and General Evaluation—is associated with its respective dataset and metrics, offering a clear summary of the experimental configuration prior to discussing detailed results.

Factual Knowledge Probing

Fig 4 shows the results of the Factual Knowledge Probing experiment in the study by \cite{kang2023impact}, which investigates the factual knowledge of LLMs using the LAMA-TREx dataset. The sentences used for validation are represented as subject-relation-object triples and converted into natural language using a predefined template. For example, the triple `Texas'-'capital'-'Austin' is converted to "The capital of Texas is Austin." Each fact masks the object and is converted into a Cloze statement (e.g., "The capital of Texas is [MASK]").

We trained the model for 3 epochs on 4 RTX 3090 GPUs. The batch size per device was 32, giving a total batch size of 128. The learning rate was 2e-5, and the Adam optimizer was used with beta_1 = 0.9 and beta_2 = 0.999.

For fine-tuning, the input prompt follows the format "### Input:\n {X\} \n\ n### Response:", where X is a masked sentence. For instance, "Hydatius has the position of [MASK]." The model is supervised to predict "bishop," which is the expected answer. Details are provided in S2 File.

The factual knowledge dataset contains 20,587 samples. We used 10,294 original sentences and 10,294 counterfactual samples as random samples to train MAGNET. To evaluate the quality of counterfactuals, we computed Self-BLEU, which measures sentence similarity. The score of 0.4668 indicates moderate diversity, showing that the generated sentences are sufficiently varied while remaining natural and coherent. This balance is important for effective fine-tuning.

For evaluation, we used Hits@1. It is 1 if the correct answer is ranked first among predicted candidates, and 0 otherwise. Because LLMs are not specifically trained for factual knowledge probing, we tested three restricted output vocabularies: (1) remove stopwords, (2) gold objects, and (3) gold objects (relation-wise). The first excludes NLTK 3.8.1 stopwords. The second restricts candidates to gold objects in the entire dataset, while the third restricts them per relation.

Fig 5 shows Hits@1 under a zero-shot setting with limited candidate sets. MAGNET improves the score by ~0.12 for the largest model and by 0.13 for the 1.3B model.

Table 2 compares GPT-Neo 2.7B performance under different training strategies. The Baseline model does not address subject-object co-occurrence biases, resulting in moderate Hits@1 scores. Undersampling removes biased samples, reducing training data and diversity. This increases overfitting risk and lowers generalization, especially in Gold Objects and Relation-wise evaluations. In contrast, MAGNET generates counterfactual samples that negate frequent object associations while preserving subjects. Learning from both original and counterfactual data maintains diversity and improves generalization, yielding substantially higher Hits@1 scores across all evaluation scenarios.

Correlation Analysis

We analyzed co-occurrence statistics in the Pile dataset [48], a pre-training dataset for GPT-Neo, and correlated them with LLMs' ability to probe factual knowledge. Entities with uncountable co-occurrence counts or consisting of more than three tokens (less than 6% of all entities) were excluded. We then computed correlations for (1) zero-shot, (2) fine-tuning alone, and (3) fine-tuning using MAGNET.

Fig 6 illustrates the number of samples in each joint subject-object frequency bin, organized according to the subject frequency bin.

Fig 7 shows co-occurrence correlations in the zero-shot setting. Hits@1 scores increase linearly with subject frequency up to approximately 10^4-10^5 for the joint subject-object frequency. However, for high-frequency subjects with relatively rare object occurrences, Hits@1 drops sharply. This indicates that LLMs struggle to predict rare facts due to co-occurrence bias.

Fig 8 presents co-occurrence correlations for fine-tuning and MAGNET. Fine-tuning roughly doubles Hits@1 compared to zero-shot but still shows sharp drops for rare facts. MAGNET, in contrast, improves overall performance by approximately three times over zero-shot and shows a slower performance decline, even for rare subject-object pairs. For bins with subject frequency 10^3-10^4 and joint frequency 10^1-10^2, MAGNET demonstrates about threefold robustness to co-occurrence bias compared to zero-shot and slower decline than standard fine-tuning.

Without MAGNET, GPT-Neo 2.7B produced 5,154 correct and 3,670 incorrect answers out of 8,824. With MAGNET, the model achieved 6,177 correct and 2,647 incorrect answers. This means 1,521 predictions changed from incorrect to correct, while 498 changed from correct to incorrect.

Among the 3,670 errors without MAGNET, 989 cases involved predictions of words with higher co-occurrence counts than the ground truth. MAGNET corrected 311 of these bias-induced errors (examples in Table 3). Conversely, of the 498 predictions that changed from correct to incorrect, 382 had higher conditional probabilities under the base model, indicating that MAGNET occasionally flipped answers despite the model’s original preference for the correct option (examples in Table 4).

Overall, MAGNET effectively corrects bias-induced errors, though it occasionally flips correct answers to incorrect ones. These cases typically occur when the object distribution for a subject is relatively uniform, meaning no single object dominates co-occurrence statistics. As a future direction, constraining counterfactual generation to subjects with strongly skewed object distributions could reduce unnecessary flips and further improve model performance.

Results for Open LLM

We evaluated the impact of MAGNET on the target models across multiple benchmarks. In addition to TruthfulQA, we included HellaSwag and Winogrande, with results summarized in Table 5. For TruthfulQA, we used MC2 (Multi-true), which computes the normalized probability assigned to the correct answer set given multiple true/false options. HellaSwag and Winogrande were evaluated using multiple-choice accuracy, representing the proportion of correct selections among four candidate continuations and pronoun disambiguation questions, respectively.

Models were trained on 4 RTX 3090 GPUs for 3 epochs, using a batch size of 256 and a learning rate of 2e-5. The Adam optimizer was employed with beta_1 = 0.9 and beta_2 = 0.999. All other procedures follow HuggingFace's causal language modeling scripts [58]. Details of fine-tuning and evaluation are provided in S3 File.

We further investigated the effect of training data size on GPT-Neo 125M using MAGNET, as shown in Fig 9. These experiments were single runs.

Overall, MAGNET effectively mitigates co-occurrence bias, reducing the likelihood of generating incorrect words with high co-occurrence probability. This improves the factual accuracy and truthfulness of model outputs.

Ablation Study

To evaluate the quality of counterfactual sentences, we conducted generation experiments using 1-shot, 5-shot, and 10-shot prompt settings. Results are summarized in Table 6.

As Table 6 shows, increasing the number of shots improves the model’s ability to follow the prompt and produce correctly formatted outputs, enabling more effective data collection. The 10-shot setting consistently yielded the highest performance across benchmarks.

To ensure the factual accuracy of generated sentences, we performed human filtering to verify that the subject and object were preserved. Table 7 compares performance with and without this filtering.

Human filtering consistently improved performance, confirming that maintaining subject-object fidelity and factual consistency enhances model outputs.

We also compared counterfactual generation using GPT-3.5 Turbo and GPT-4 Turbo. Table 8 summarizes the results.

As shown, GPT-4 Turbo consistently outperformed GPT-3.5 Turbo across all benchmarks. These results emphasize the importance of generating high-quality, factually grounded counterfactual sentences while preserving the subject-object structure. They also indicate that using a sufficient number of in-context shots and leveraging more advanced LLMs can further enhance MAGNET’s effectiveness, improving both the factual robustness and truthfulness of the target models.

Reviewer #1 - Comment 2

Comment Some figure captions (especially Figures 7–9) could be made more concise, and figure quality should be upgraded to ensure all text and visual elements are clearly visible.

Response We thank the reviewer for the helpful suggestion regarding figure captions and quality. We have revised the captions for Figures 1, 2, 3, 4, 5, 7, 8, and 9 to make them more concise, and increased the text size in all figures to ensure that all labels and visual elements are clearly visible.

[Page 4](Figure 1) – Section Methods

[Page 5](Figure 2) – Section Methods

[Page 6](Figure 3) – Section Methods

[Page 6](Figure 4) – Section Results

[Page 7](Figure 5) – Section Results

[Page 8](Figure 7) – Section Results

[Page 8](Figure 8) – Section Results

[Page 9](Figure 9) – Section Results

Reviewer #1 - Comment 3

Comment Including a short summary table in the Methods section listing datasets, benchmarks, and evaluation metrics would improve readability.

Response We thank the reviewer for this helpful suggestion. Because all datasets, benchmarks, and evaluation metrics are introduced in the Results section, we added a summary table in that section along with a concise explanatory paragraph to improve readability and provide a clear overview of the experimental setup.

[Page 6](Line 181–185) – Section Results

Results

Table 1 provides a concise overview of the datasets, benchmarks, and evaluation metrics used in our experiments. Each experiment category—Factual Knowledge Probing, Counterfactual Training (MAGNET), Bias Analysis, and General Evaluation—is associated with its respective dataset and metrics, offering a clear summary of the experimental configuration prior to discussing detailed results.

[Page 6](Table 1) – Section Results

Overview of datasets, benchmarks, and corresponding evaluation metrics for each experiment category.

Reviewer #1 - Comment 4

Comment A light grammar and style edit is advised to smooth a few long sentences in the Introduction and Results sections.

Response We appreciate the reviewer’s suggestion regarding grammar and style. In response, we carefully revised both the Introduction and Results sections to smooth overly long sentences and improve clarity and readability throughout the manuscript.

[Page 1-2](Line 2-30) – Section Introduction

Introduction

Natural language processing (NLP) research has recently experienced rapid growth with the emergence of large language models (LLMs) [1,2]. LLMs have demonstrated strong performance across a wide range of NLP tasks, including natural language inference [3], question answering [4], common-sense reasoning [5], and translation [6]. They have also achieved significant gains in natural language generation tasks. However, the problem of hallucination—the generation of plausible but untruthful sentences—has attracted considerable attention. Early work focused on the likelihood-maximizing objective function used during training and decoding, showing that natural language generation models can produce sentences that are plausible yet nonsensical or untruthful [7,8].

Recent studies suggest that LLMs often learn spurious features, which can lead to untruthful sentences [9]. Inspired by [10], we identify co-occurrence statistics in pre-trained sentences as a major contributor to these spurious features. Kang et al. proposed a fine-tuning method that removes biased samples from the dataset. While this approach mitigates hallucination caused by high co-occurrence statistics, it can hurt generalization due to the reduced data size.

In this paper, we propose MAGNET (Model-AGNostic coutErfacTual synthesis and adaptive fine-tuning), a framework designed to address bias in fine-tuning datasets by generating counterfactual samples for all instances, rather than removing biased samples. Counterfactual samples have been widely used in NLP to mitigate spurious features such as co-occurrence bias [11], and several studies have leveraged them for data augmentation [12-15]. Most methods generate counterfactuals by identifying and replacing terms that play a crucial role in a sentence’s causality.

Using MAGNET presents two main challenges. First, generating counterfactuals to address subject-object co-occurrence bias requires extracting the subject and object, typically using part-of-speech (POS) tagging. In our approach, we directly utilize the subject and object information provided by LAMA-TREx. Second, counterfactual sentences should retain the subject while negating the object. This task requires broad knowledge and common-sense reasoning. To address this, we leverage GPT-3’s powerful few-shot learning ability to generate counterfactual sentences effectively.

[Page 6-11](Line 181-297) – Section Results

Results

Table 1 provides a concise overview of the datasets, benchmarks, and evaluation metrics used in our experiments. Each experiment category—Factual Knowledge Probing, Counterfactual Training (MAGNET), Bias Analysis, and General Evaluation—is associated with its respective dataset and metrics, offering a clear summary of the experimental configuration prior to discussing detailed results.

Factual Knowledge Probing

Fig 4 shows the results of the Factual Knowledge Probing experiment in the study by \cite{kang2023impact}, which investigates the factual knowledge of LLMs using the LAMA-TREx dataset. The sentences used for validation are represented as subject-relation-object triples and converted into natural language using a predefined template. For example, the triple `Texas'-'capital'-'Austin' is converted to "The capital of Texas is Austin." Each fact masks the object and is converted into a Cloze statement (e.g., "The capital of Texas is [MASK]").

We trained the model for 3 epochs on 4 RTX 3090 GPUs. The batch size per device was 32, giving a total batch size of 128. The learning rate was 2e-5, and the Adam optimizer was used with beta_1 = 0.9 and beta_2 = 0.999.

For fine-tuning, the input prompt follows the format "### Input:\n {X\} \n\ n### Response:", where X is a masked sentence. For instance, "Hydatius has the position of [MASK]." The model is supervised to predict "bishop," which is the expected answer. Details are provided in S2 File.

The factual knowledge dataset contains 20,587 samples. We used 10,294 original sentences and 10,294 counterfactual samples as random samples to train MAGNET. To evaluate the quality of counterfactuals, we computed Self-BLEU, which measures sentence similarity. The score of 0.4668 indicates moderate diversity, showing that the generated sentences are sufficiently varied while remaining natural and coherent. This balance is important for effective fine-tuning.

---

## [Decision Letter · Decision Letter 2]

28 Dec 2025

MAGNET: Counterfactual samples synthesizing for mitigating hallucination in large language models

PONE-D-25-17618R2

Dear Dr. Jang,

We’re pleased to inform you that your manuscript has been judged scientifically suitable for publication and will be formally accepted for publication once it meets all outstanding technical requirements.

Kind regards,

Sonia Vasconcelos, PhD

Academic Editor

PLOS One

Reviewers' comments:

Reviewer's Responses to Questions

**Comments to the Author**

1. If the authors have adequately addressed your comments raised in a previous round of review and you feel that this manuscript is now acceptable for publication, you may indicate that here to bypass the “Comments to the Author” section, enter your conflict of interest statement in the “Confidential to Editor” section, and submit your "Accept" recommendation.

Reviewer #1: All comments have been addressed

2. Is the manuscript technically sound, and do the data support the conclusions?

Reviewer #1: Yes

3. Has the statistical analysis been performed appropriately and rigorously?

Reviewer #1: Yes

4. Have the authors made all data underlying the findings in their manuscript fully available?

Reviewer #1: Yes

5. Is the manuscript presented in an intelligible fashion and written in standard English?

Reviewer #1: Yes

6. Review Comments to the Author

Reviewer #1: The authors have satisfactorily addressed all comments from previous review rounds. The manuscript is technically sound, clearly written, and supported by appropriate experimental design and statistical analysis. Data availability requirements are fully met. Overall, the study meets the standards of PLOS ONE and is suitable for publication.

7. PLOS authors have the option to publish the peer review history of their article (what does this mean?). If published, this will include your full peer review and any attached files.

Reviewer #1: No

---

## [Editor Report · Acceptance letter]

PONE-D-25-17618R2

PLOS One

Dear Dr. Jang,

I'm pleased to inform you that your manuscript has been deemed suitable for publication in PLOS One. Congratulations! Your manuscript is now being handed over to our production team.

Kind regards,

on behalf of

Dr. Sonia Vasconcelos

Academic Editor

PLOS One